# Pulmonary-Renal Syndrome from Levamisole-Adulterated Cocaine-Induced Antineutrophil Cytoplasmic Antibody (ANCA)-Associated Vasculitis: A Systematic Review

**DOI:** 10.3390/ph16060846

**Published:** 2023-06-06

**Authors:** Philip Bucur, Marshall Weber, Rashi Agrawal, Adria Irina Madera-Acosta, Rachel E. Elam

**Affiliations:** 1Department of Medicine, Medical College of Georgia, Augusta University, Augusta, GA 30912, USA; 2Charlie Norwood Veterans Affairs Medical Center, Augusta, GA 30901, USA; 3Medical College of Georgia, Augusta University, Augusta, GA 30912, USA; (M.W.);; 4Mayo Clinic Health System, Luther Campus Clinic, Eau Claire, WI 54703, USA; 5Division of Rheumatology, Department of Medicine, Medical College of Georgia, Augusta University, Augusta, GA 30912, USA

**Keywords:** vasculitis, levamisole, pulmonary-renal syndrome, cocaine, diffuse alveolar hemorrhage, glomerulonephritis

## Abstract

Levamisole is an anti-helminthic drug with immunomodulatory properties that is added to cocaine to increase its potency and weight. Levamisole-adulterated cocaine (LAC) may cause an antineutrophil cytoplasmic antibody (ANCA)-associated systemic small vessel vasculitis (AAV). We aimed to characterize the phenotype of persons developing pulmonary-renal syndrome (PRS) in LAC-induced AAV and summarize its treatment and outcomes. Pubmed and Web of Science were searched (until September 2022). Reports that described co-existing diffuse alveolar hemorrhage and glomerulonephritis in an adult (age ≥ 18) with confirmed or suspected LAC exposure were included. Reports, demographics, clinical and serologic features, treatment and outcome characteristics were extracted. Of the 280 records identified, eight met the inclusion criteria, including eight unique cases. Persons were aged 22–58 years, and 50% were women. Cutaneous involvement occurred in only half of the cases. Other associated vasculitis findings and serologies were heterogeneous. All patients received immunosuppression with steroids, with cyclophosphamide and rituximab commonly added. We concluded that PRS could occur from LAC-induced AAV. Distinguishing LAC-induced AAV from primary AAV is challenging as clinical and serologic presentations overlap. Asking about cocaine use is requisite in persons presenting with PRS to guide diagnosis and appropriately counsel on cocaine cessation in conjunction with immunosuppression as treatment.

## 1. Introduction

Levamisole is an anti-helminthic drug with immunomodulatory properties that was previously used in the treatment of various medical conditions, including autoimmune disorders and as an adjuvant chemotherapy agent for cancer [1]. The therapeutic use of levamisole in humans has been discontinued in many countries due to severe adverse effects, including agranulocytosis, seizures, skin necrosis, and vasculitis [2,3,4]. A systemic, antineutrophil cytoplasmic antibody (ANCA)-positive vasculitis has been reported in patients on therapeutic doses of levamisole [5,6,7]. Levamisole remains widely available as a veterinary medicine and is commonly used as a cutting agent for cocaine to increase its potency and weight [3,8]. In 2019, the crude global prevalence of cocaine use disorder was 0.06% (95% confidence interval, 0.04–0.07%) [9]. Therefore, an estimated 308 to 539 million persons globally are regularly using cocaine [9,10]. The prevalence of cocaine use disorder is highest in the Americas and Europe [9]. In these high-use regions, levamisole has been detected as an adulterant in 52–69% of cocaine samples systematically surveyed [11,12], and 88% of urine samples positive for cocaine are also positive for levamisole in one study in San Francisco, California, USA [13]. As such, levamisole-adulterated cocaine-induced (LAC-induced) vasculitis is an important global public health concern [3].

The typical presentation of LAC-induced vasculitis is inflammation of small vessels associated with necrosis and disruption of the vessel walls, leading to retiform purpura with or without hemorrhagic bullae and/or skin necrosis (especially common on the ear helixes and cheeks), arthralgia, otolaryngologic involvement (sinusitis, rhinorrhea, ulcers, and midline destruction of the nasal septum) and constitutional symptoms [3,14,15]. The majority of these cases demonstrate autoantibodies, including ANCAs, anti-phospholipid antibodies [16], antinuclear antibodies (ANA), anti-double-stranded DNA (dsDNA), anti-human neutrophil elastase (HNE) antibody, and cryoglobulins [3,4,14,15]. Cases positive for ANCA may demonstrate myeloperoxidase [17] antibody or proteinase-3 (PR3) antibody, both, or neither, and may be termed LAC-induced ANCA-associated vasculitis (AAV) [14,15].

The habitual use of cocaine may cause complications affecting virtually any organ system [18], and there is substantial overlap in the clinical presentations of cocaine abuse alone and LAC-induced AAV [14,19]. The medical complications seen in cocaine abuse may depend on the route of cocaine administration [18]. With chronic nasal insufflation, cocaine users may develop progressive mucosal and perichondrial injury and subsequent ischemic necrosis directly from cocaine exposure, termed cocaine-induced midline destructive lesion (CIMDL). Often presenting with nasal obstruction, crusting, epistaxis, septal and/or palatal perforation, severe sinusitis and facial pain, CIMDL closely mimics the otolaryngologic involvement of LAC-induced AAV and granulomatosis with polyangiitis both clinically and histopathologically [14,19].

In contrast, smoking cocaine leads to a myriad of lower respiratory tract manifestations, including acute respiratory symptoms (cough, chest pain, hemoptysis), exacerbation of asthma, thermal airway injury, deterioration in lung function, pneumothorax, pneumomediastinum, bronchiolitis obliterans with organizing pneumonia, noncardiogenic pulmonary edema, interstitial pneumonitis, pulmonary hypertension and diffuse alveolar hemorrhage (DAH) [20]. DAH, as a consequence of intense cocaine-induced vasoconstriction, may be clinically indistinguishable from DAH secondary to vasculitis in the absence of a lung biopsy [21]. Cocaine use may induce renal injury via rhabdomyolysis, malignant hypertension, accelerated hypertensive nephrosclerosis, renal artery thrombosis or dissection, acute interstitial nephritis, and thrombotic microangiopathy [22]. When cocaine is adulterated with levamisole, the development of pauci-immune crescentic glomerulonephritis is further reported [22].

Pulmonary-renal syndrome (PRS) is a potentially life-threatening condition defined by the combination of DAH and rapidly progressive glomerulonephritis (RPGN) [23]. While direct toxicity of cocaine may cause DAH, the underlying cause of PRS is almost always systemic small-vessel vasculitis. By far, the most common vasculitis is primary AAV (i.e., granulomatosis with polyangiitis, eosinophilic granulomatosis with polyangiitis, and microscopic polyangiitis), accounting for approximately 56–77.5% of cases of PRS [24].

Acute renal failure and alveolar hemorrhage independently are each well-described but uncommon manifestations of LAC-induced vasculitis, affecting approximately 12.5% and 3.1% of cases, respectively [14,15]. Causes of acute renal failure in LAC-induced vasculitis are manifold, including crescentic pauci-immune glomerulonephritis (as is classic in primary AAV), but also crescentic immune-complex mediated glomerulonephritis, membranous nephropathy, and thrombotic microangiopathy [15]. The simultaneous occurrence of DAH and RPGN as a combined PRS in LAC-induced AAV is exceedingly rare [25,26,27,28,29,30,31,32] and, as such, may go unrecognized and be treated as primary AAV by many rheumatologists and other clinicians. However, existing treatment paradigms for LAC-induced AAV and primary AAV differ; the mainstay of treatment for the former is the discontinuation of cocaine use, while the latter is treated with systemic immunosuppression, typically including glucocorticoids, a choice of rituximab or cyclophosphamide, and consideration of the addition of the C5a receptor inhibitor, avacopan [4,15,33,34].

The purpose of this literature review was two-fold: [1] to characterize the phenotype of persons who develop PRS secondary to LAC-induced AAV, and [2] to summarize the treatments employed and their outcomes in known cases of PRS secondary to LAC-induced AAV.

## 2. Methods

Two primary databases (PubMed and Web of Science) were searched from inception until 2 September 2022 using the following keywords: (“levamisole” OR “cocaine”) AND “pulmonary-renal syndrome” OR “glomerulonephritis” OR “alveolar hemorrhage” OR (“vasculitis” AND “kidney”) OR (“vasculitis” AND “lung”) OR (“vasculitis” AND “pulmonary”) OR (“vasculitis” AND “renal”). The search was limited to studies written in English and involving human subjects. Additional citations were added through a Google Scholar search and review of references from included reports. Case reports and case series were eligible for inclusion if they [1] reported at least 1 case of pulmonary-renal syndrome (PRS; defined as co-existing DAH and glomerulonephritis) and the case of interest was reported in [2] an adult (aged at least 18 years) with [3] confirmed or suspected levamisole-adulterated cocaine (LAC) exposure, and [4] a positive ANCA serologic test. Studies were excluded if they reported a case [1] in a pediatric patient (aged < 18 years), [2] the case only described pulmonary or renal vasculitis manifestations, but not both occurring simultaneously, or [3] there was insufficient evidence reported to confidently conclude the presence of DAH or glomerulonephritis as the etiology of the pulmonary or renal involvement, respectively.

One author (P.B., M.W., or R.A.) individually screened each article (title, keywords and abstract), and subsequently, the full texts of records recognized as potentially eligible by abstract screening. A second reviewer and subject matter expert (R.E.E.) was included to resolve discrepancies. Predetermined fields for data extraction from each report included demographics (age, sex, race, ethnicity), the time between onset/diagnosis of vasculitis and onset of PRS manifestation, the method of ascertainment of levamisole exposure, evidence of DAH, evidence of glomerulonephritis, cutaneous manifestations of vasculitis (if present), other organ system manifestations of vasculitis (if present), ANCA profile and other serologies, renal indices, treatment(s) described, and outcome(s) of the case.

This study was reviewed by the Augusta University Institutional Review Board and determined not to be research involving human subjects.

## 3. Results

Two hundred eighty records were identified from the two primary databases. Thirty-five additional records were identified through a Google Scholar search and review of references from included reports. Of these records, eight met the inclusion criteria. A flow chart of the literature search strategy is depicted in Figure 1. Each included article identified detailed one unique case of PRS secondary to LAC-induced AAV. Six articles were full-text manuscripts, and two were published abstracts. The characteristics of the eight identified cases of PRS secondary to LAC-induced AAV are summarized in Table 1. Serologic profiles and renal indices at the time of PRS presentation in the included cases are detailed in Table 2.

### 3.1. Patient Characteristics

Age ranged from 22 to 58 years with a mean age of 41 years (standard deviation 11.7 years). Half of the cases occurred in men and half in women. Race and ethnicity were reported very infrequently. The majority of cases (63%) presented with PRS as the initial manifestation of LAC-induced AAV, but others reported PRS onset after a long-standing LAC-induced AAV diagnosis. Two of the eight cases reported a pre-existing autoimmune diagnosis in the past medical history: one patient had a history of reactive arthritis [31], and another had a history of systemic lupus erythematosus [32]. Half of the cases determined levamisole exposure based on cocaine use by history, and the other half of cases confirmed cocaine positivity on urine drug screening. Only one case further confirmed levamisole in a urine sample by liquid chromatography-mass spectrometry [27].

### 3.2. Clinical and Pathologic Features

Most cases (75%) confirmed DAH with bronchoscopy, but two cases diagnosed DAH based on hemoptysis and compatible chest imaging without bronchoscopy. Renal biopsy confirmed glomerulonephritis in seven cases (88%). The remaining case was suggestive of glomerulonephritis by reporting the presence of hematuria and proteinuria on urinalysis [29]. Of cases with renal biopsy, rapidly progressive renal disease was supported by the presence of either necrosis or crescents in six of seven cases (86%). Immunofluorescence of renal biopsies was variable, with two cases reporting pauci-immune RPGN and two cases reporting immune-complex mediated disease. Two cases reported concomitant membranous nephropathy.

The most commonly reported associated organ involvement was otolaryngologic (63%), with epistaxis and destruction/perforation of nasal septum reported. Cutaneous and musculoskeletal involvement each occurred in 50% of cases. Two cases (25%) had both arthralgia and skin lesions consisting of erythema to purpura/necrosis on the face and extremities [25,29]. The most common skin biopsy finding reported was leukocytoclastic vasculitis. Constitutional symptoms were found in three cases (38%), and each of the following vasculitis features was reported once: testicular vasculitis, venous thrombosis, and heart failure. One case had no reported associated vasculitis features outside of pulmonary and renal involvement as PRS.

### 3.3. Serologic and Other Laboratory Findings

The majority of cases were positive for myeloperoxidase [17] antibody (63%). When the ANCA pattern was reported, these patients with positive MPO antibodies demonstrated typical perinuclear ANCA (P-ANCA) positivity as well (three of five cases). Two cases were proteinase-3 (PR3) antibody positive (25%). One case was PR3 positive in the setting of high titer cytoplasmic ANCA (C-ANCA) positivity, negative P-ANCA and negative MPO antibody [30]. The other was dual PR3 and MPO antibody positive [31]. One case was low-titer positive for P-ANCA (1:80) and negative for C-ANCA, PR3, and MPO. Positive APLA, ANA, dsDNA, rheumatoid factor (RF), Anti-Ro, and Anti-La were all reported, but none consistently across more than two studies. In one published abstract, ANCA serologies were not directly reported, but this patient had a history of LAC-induced AAV, and the authors state, “workup suggested a recurrence of [LAC-induced AAV] in the setting of recent cocaine use” [32].

Creatinine at presentation with PRS varied widely from normal to very high (up to 11.07 mg/dL). When urinalysis results were reported in six of eight cases, both proteinuria and hematuria were present. Half of these cases reported nephrotic range proteinuria, one case reported non-nephrotic range proteinuria, and the remaining two cases did not specify the amount of proteinuria. Dysmorphic and crenated red blood cells were reported on urine microscopy.

### 3.4. Treatment and Outcomes

All eight of the cases reported were treated with high-dose steroids. Six of eight cases received additional immunosuppression with cyclophosphamide (63%) and/or rituximab (25%). One person who initially received IV cyclophosphamide was transitioned to maintenance azathioprine therapy but did not tolerate this medication [27]. The maintenance of the immunosuppressive therapy regimen was not reported in any other case. Three persons required hemodialysis (38%), two patients received plasmapheresis (25%), and one person received IV recombinant activated factor VII (rFVIIa) for refractory DAH. Two reported cases of PRS in LAC-induced AAV remained dialysis-dependent, but the remaining cases reported at least partial recovery. The third case reporting hemodialysis as an acute intervention did not report the renal outcome.

## 4. Discussion

Pulmonary-renal syndrome (PRS) can occur from levamisole-adulterated cocaine-induced antineutrophil cytoplasmic antibody-associated vasculitis (LAC-induced AAV). In this literature review, we determined that PRS from LAC-induced AAV is gender indiscriminate and highly heterogeneous in phenotype with regards to age, associated clinical features and analytical results, which may lead to diagnostic and treatment difficulties.

The diagnosis of LAC-induced AAV is a diagnosis of exclusion based on compatible clinical presentation in the context of a reasonable ascertainment of exposure to levamisole-adulterated cocaine. The classic presentation of LAC-induced AAV includes fever, arthralgia, otolaryngologic symptoms, and, in the vast majority of cases, cutaneous involvement, most commonly as retiform purpuric lesions with or without central necrosis, with a special predilection for the face and earlobes [14,15,35]. Typical pathologic lesions on skin biopsy are leukocytoclastic vasculitis and/or thrombotic vasculopathy with the involvement of superficial and deep dermal vessels [35,36]. In this review, these usual findings of LAC-induced AAV that may prompt the rheumatologist or other clinician to prioritize a LAC-induced AAV diagnosis as the potential cause of PRS were not consistently described. While skin involvement is seen in over 90% of cases of LAC-induced vasculitis [15,37], only 50% of cases of PRS secondary to LAC-induced vasculopathy in this review reported any associated cutaneous lesions. Further, PRS occurred as the presenting symptom of LAC-induced AAV in the majority of the cases described. Clinical symptoms alone are unlikely to help differentiate PRS from LAC-induced AAV versus primary and other secondary forms of AAV.

Antineutrophil cytoplasmic antibodies (ANCA), autoantibodies directed against antigens in the cytoplasmic granules of neutrophils and monocytes, may be helpful in distinguishing types of ANCA-associated vasculitis (AAV) with often overlapping clinical presentations [38]. Granulomatosis with polyangiitis typically displays a pattern of cytoplasmic ANCA (C-ANCA) almost exclusively as a result of antibodies directed against proteinase-3 (PR3), while microscopic polyangiitis typically shows a pattern of perinuclear ANCA (P-ANCA) targeting myeloperoxidase [17,38]. The ANCA pattern that characterizes LAC-induced AAV is a positive P-ANCA with or without positive MPO titers [14,36]. A potential discordance between P-ANCA and MPO in some cases of LAC-induced AAV has been attributed to P-ANCA production directed at other neutrophil granule components such as human neutrophil elastase (HNE), lactoferrin, cathepsin G, and sometimes PR3 [36,38,39,40]. P-ANCA targeting HNE is reported in cocaine-induced midline destructive lesions and LAC-induced cutaneous vasculopathy but is notably absent in primary AAV [40]. Combined positivity for MPO antibody and PR3 antibody is reported in drug-induced AAV, including LAC-induced AAV, but is uncommon in primary AAV [14,15,41]. HNE and cathepsin G share epitopes with PR3, which may also be responsible for the combined positivity for MPO and PR3 antibodies through cross-reactivity in LAC-induced AAV [26]. In addition, patients with LAC-induced AAV are often positive for other autoantigens, including ANA, APLA, anti-dsDNA, and others [3,4,14,15].

There are limited reports of levamisole-induced AAV in the absence of concomitant cocaine exposure [5,6,7,35,42], and none, to our knowledge, report pulmonary or renal vasculitis secondary to isolated levamisole exposure. The exact pathogenesis of LAC-induced AAV is unclear, but several potential mechanisms have been described. Levamisole has been shown to cause immune system dysregulation that could provoke an autoimmune vasculitis in genetically predisposed individuals through various mechanisms: formation of antibodies with increased B and T cells, chemotaxis and increased neutrophil response, deposition of immune complexes in blood vessel walls, and degranulation and release of oxygen metabolites leading to vascular injury [35]. Importantly, both cocaine and levamisole induce the formation of neutrophil extracellular traps (NETs) enriched in neutrophil elastase, a potential target for ANCAs and a mechanism for loss of self-tolerance in individuals who develop LAC-induced AAV [43]. Both cocaine and levamisole also increase the release of B-cell activating factor (BAFF), which may promote the survival and differentiation of B cells producing pathologic ANCAs in LAC-induced AAV [43].

In this review, our findings in patients with PRS secondary to LAC-induced AAV were consistent with existing literature regarding serologic profiles of LAC-induced AAV in general. However, the serologic profiles reported were non-uniform. The most commonly seen ANCA serologic profile was P-ANCA positivity without or without MPO antibody; however, this accounted for only half of the cases. Dual PR3 and MPO antibody positivity was reported in only one case. One case displayed typical C-ANCA targeting PR3-positivity indistinguishable from the typical ANCA profile of granulomatosis with polyangiitis. Positive APLA, ANA, dsDNA, RF, Anti-Ro, and Anti-La were all reported, but none consistently across more than two studies. No serologic features predominated in this review that would consistently distinguish PRS secondary to LAC-induced AAV from PRS secondary to primary AAV. Anti-HNE was not reported in any of the cases in our review. Lood and Hughes have demonstrated that the pathogenesis of LAC-induced AAV may be driven by the ability of cocaine and levamisole to induce the release of inflammatory neutrophil extracellular traps (NETs), leading to neutrophil elastase (HNE) autoantigen production [43]. Systematic ascertainment of anti-HNE may be an important future direction that deserves clinical attention in future series of LAC-induced AAV with PRS as this could be a means of distinguishing LAC-induced AAV from primary AAV.

Levamisole was confirmed in the urine by liquid chromatography-mass spectrometry (LC-MS) in only one of eight cases of PRS due to LAC-induced AAV in this review. Instead, cocaine exposure was determined either by history alone or with a conventional urine drug screen. The pharmacokinetics of levamisole, with a short plasma elimination half-life of 5.6 h, extensive metabolism, and minimal unchanged drug urinary excretion (2–5%), make it a difficult agent to detect in biological samples [44,45]. Clinicians treating patients presenting with PRS must have a high index of suspicion for LAC-induced AAV early as the window for confirmatory testing for urinary levamisole with LC-MS is very narrow. However, given the high rates of adulteration of cocaine with levamisole [11,12,13], confirmation of levamisole in urine samples is likely unnecessary if a thorough and detailed social history is undertaken to ascertain a history of regular cocaine exposure when possible. When history is limited due to intubation or otherwise, a conventional urinary drug screen positive for cocaine would still be highly suggestive of levamisole-adulterated cocaine exposure.

More information is needed about the optimal treatment of PRS secondary to LAC-induced AAV. The existing paradigm of treatment for LAC-induced AAV, in general, has been a complete cessation of cocaine (and thereby, levamisole) use [14,35,46]. In the absence of internal organ involvement, additional immunosuppression may be unnecessary, although relapse rates are high with the reintroduction of cocaine [15,37]. Corticosteroids are commonly used in life and/or organ-threatening LAC-induced AAV and were given to all patients with a manifestation of PRS in this review [15,26,37]. Cyclophosphamide or rituximab are the most commonly reported additional immunosuppressants used to treat PRS secondary to LAC-induced vasculitis, which is in accordance with existing clinical practice guidelines for the treatment of primary AAV with active severe (life- or organ-threatening manifestations) disease [33] and existing reports for other severe LAC-induced vasculitis cases [15,26]. Rituxan is a much newer agent compared to cyclophosphamide to treat adult patients with granulomatosis with polyangiitis and microscopic polyangiitis in adults in combination with glucocorticoids (U.S. Food and Drug Administration approved since 2011) [47], which may or may not account for more use of cyclophosphamide in this review.

Plasmapheresis was used twice in these cases, once with reported recovery in a patient with the primary manifestation of DAH over renal involvement and once in a patient who required hemodialysis and remained dialysis-dependent at the time of publication. In both instances, the cases occurred before the PEXIVAS trial in 2020 demonstrated no reduction in death or kidney failure with the use of plasmapheresis in persons with severe primary AAV [48]. However, in our review, immune complex deposition was more regularly reported on skin and renal biopsies than in classic primary AAV, so the role of plasmapheresis in LAC-induced AAV remains unclear. For refractory DAH, IV recombinant activated factor VII was used in one case with a favorable outcome. There is a paucity of data on maintenance therapy for PRS secondary to LAC-induced AAV. Only a single case in this review reported on any maintenance therapy being employed, and the agent used was azathioprine, with no data on the outcome of its use as the patient did not tolerate the azathioprine [27]. Dialysis dependence is unfortunately not uncommon in patients with PRS from LAC-induced AAV per this review, as has also been reported in other cases of isolated RPGN from LAC-induced AAV [26,49,50].

This review has limitations. The data on evidence for DAH, evidence for glomerulonephritis, or associated vasculitis symptoms may have been incompletely reported in manuscripts or published abstracts, leading to incomplete case ascertainment or lack of detail to inform clinical phenotypes in this review. Data on ANCA profiles was incomplete in several included published reports which may have impaired our ability to characterize the serologic profile of PRS in LAC-induced AAV. We were unable to distinguish in each case with otolaryngologic involvement whether these manifestations were due to LAC-induced AAV or direct toxicity of cocaine, i.e., CIMDL, given their substantial overlap in clinical presentation, as is often the case in clinical practice. With the exception of Pecci et al. [29], where bronchial biopsy confirmed pulmonary vasculitis, we were unable to definitively attribute DAH in these cases to LAC-induced AAV rather than a direct toxic effect of smoked cocaine-induced intense vasoconstriction. However, the concomitant presence of RPGN reasonably suggests systemic vasculitis as the etiology of PRS in these cases, as RPGN is not a reported feature of cocaine-induced pathology in the absence of vasculitis [22]. Levamisole exposure was not confirmed by LC-MS in urinary samples in most cases. Finally, publication bias may influence the phenotypes of PRS secondary to LAC-induced AAV published and subsequently described here.

## 5. Conclusions

Asking about cocaine use is requisite in persons presenting with PRS. PRS may be due to LAC-induced AAV. Differentiation of PRS secondary to LAC-induced AAV versus primary and other secondary causes of AAV can be challenging due to the heterogeneity of clinical and serologic presentations in PRS secondary to LAC-induced AAV and their substantial overlapping features with primary AAV. A detailed history eliciting any cocaine exposure may be the best and sometimes only clue to separating these entities. Recognizing the diagnosis of LAC-induced AAV is difficult but crucial. The prognosis for PRS secondary to LAC-induced AAV may depend on the patient’s understanding and willingness to discontinue cocaine use, although data on cocaine reintroduction and recurrence of PRS was beyond the scope of this review. Although PRS secondary to either LAC-induced AAV or primary AAV are treated with similar immunosuppressive regimens, it is necessary to identify LAC-induced AAV in order to appropriately counsel on cocaine cessation to improve the chance of recovery and avoid recurrence.

## Figures and Tables

**Figure 1 pharmaceuticals-16-00846-f001:**
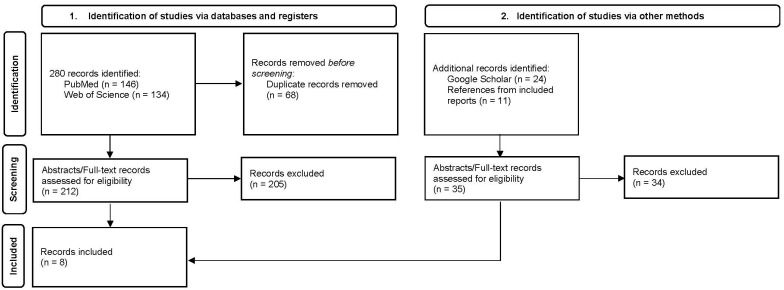
Diagram of the literature search strategy.

**Table 1 pharmaceuticals-16-00846-t001:** Characteristics of cases of pulmonary-renal syndrome secondary to levamisole-adulterated cocaine-induced systemic antineutrophil cytoplasmic antibody (ANCA)-associated vasculitis identified through literature review.

Reference	Age (Years)	Sex	Race, Ethnicity	Duration of Vasculitis Prior to PRS	Evidence for Levamisole Exposure	Evidence for DAH	Evidence for RPGN	Skin Lesions	Other Vasculitis Findings	Treatment	Outcome
Neynaber et al., 2008 [30]	22	M	NR, NR	PRS at presentation	Cocaine use x 2 years by history	CXR: dense infiltrates in the RUL and RML compatible with pneumonia and intra-alveolar hemorrhage; Bronchoscopy: no infectious agent; No response to Abx therapy and rapid response to immune-suppression	Renal bx: focal and segmental pauci-immune, crescentic RPGN	Nodules and plaques; skin bx: leukocytoclastic vasculitis of small and medium-sized dermal vessels with fibrinoid necrosis; direct IF: positive for IgM (large), and lesser amounts of IgG, IgA and C3	Destruction of nasal septum, testicular vasculitis, venous thrombosis	oral CYC, high-dose steroids	Recovery
Pecci et al., 2013 [29]	37	M	NR, NR	PRS at presentation	Cocaine use by history	CT chest: right interstitial diffuse alveolar infiltrates; Bronchoscopy: ongoing bleeding from the right bronchial tree & no infectious agent; Bronchial bx: vasculitis	Urinalysis: Hematuria, proteinuria	Purpura and necrosis of legs and earlobes	Arthralgia, malaise, perforated nasal septum	high-dose steroids, plasmapheresis, intravenous rFVIIa for refractory DAH	Recovery
Carlson et al., 2014 [26]	49	F	NR, NR	Five months (untreated renal involvement preceding PRS)	UDS: +cocaine	CXR: Bilateral pulmonary infiltrates; Bronchoscopy consistent with DAH	Renal bx: Focal segmental and global sclerosing glomerulopathy with cellular crescents, pauci-immune	Leg ulcers; skin bx: leukocytoclastic vasculitis with fibrin thrombi of superficial and deep dermal vessels	None reported	IV CYC, high-dose steroids, plasmapheresis, hemodialysis	Dialysis dependent
Collister et al., 2017 [27]	35	M	White, NR	PRS at presentation	UDS: +cocaine; LC-MS: +levamisole	CT chest: bilateral ground glass opacifications in the setting of hemoptysis and iron deficiency anemia	Renal bx: Membranous nephropathy with fibrinoid necrosis and crescent formation; IF: mesangial and capillary loop IgG (1-2+), IgM (1+), C3 (1+), trace C1q, and lambda (1+) and negative for IgA, fibrinogen, and kappa	NR	Distal symmetric polyarthritis, chronic epistaxis	IV CYC, high-dose steroids followed by steroid taper, maintenance azathioprine (did not tolerate) and ACEI	Recovery with residual CKD stage 3A
Berlioz et al., 2017 [25]	41	F	NR, Hispanic	Six years	UDS: +cocaine	CT chest: prominent bilateral patchy infiltrates and symmetric moderate pleural effusions; Bronchoscopy: confirmed DAH	Renal bx: Focal segmental necrotizing and sclerosing GN; membranous nephropathy also present	Tender skin erythema of nose, extremities; skin bx: neutrophil-rich infiltrate around the vessels of the dermis with fibrin thrombi within the superficial vessels → widespread skin bullae & necrosis	Arthralgia, Generalized weakness, new onset systolic heart failure	High-dose steroids, rituximab, then CYC (for worsening renal function)	Immune suppression complicated by a severe skin infection, but, ultimately, recovery
Habibullah, Lou, & Tsegaye, 2019 [28]	53	F	NR, NR	PRS at presentation	Cocaine use by history	CT chest: bilateral ground-glass and consolidated opacities; Bronchoscopy: confirmed DAH with increasing blood on serial aliquots	Renal bx: Advanced global and segmental glomerulosclerosis with healed, chronic, crescentic GN	NR	Fever, epistaxis	High-dose steroids, hemodialysis	Dialysis dependent
Restrepo-Escobar et al., 2020 [31]	34	M	NR, NR	PRS at presentation	Cocaine abuse by history	CXR: diffuse pulmonary infiltrates; CT chest: generalized mixed opacities bilateral lungs; Bronchoscopy: confirmed DAH with 40% hemosiderin-laden macrophages	Renal bx: Diffuse endocapillary GN with immune complex depositis and tubulointerstitial nephritis	NR	Generalized discomfort; perforated nasal septum	High-dose steroids → IV CYC (for only partial renal improvement)	Partial renal recovery
Voong & Patel, 2022 [32]	58	F	NR, NR	Existing diagnosis of LAC-induced AAV of duration not reported	UDS: +cocaine	CXR: severe pulm edema with bilateral opacities in the setting of severe hemoptysis	Renal bx: Diffuse necrotizing and crescentic GN	NR	NR	High-dose steroids, rituximab, hemodialysis	Respiratory status improved; renal status not reported

years: years; M: male; F: female; PRS: pulmonary-renal syndrome; DAH: diffuse alveolar hemorrhage; RPGN: rapidly progressive glomerulonephritis; ANCA: antineutrophil cytoplasmic antibody; Abs: antibodies; NR: not reported; CXR: chest x-ray; RUL: right upper lobe; RML: right middle lobe; Abx: antibiotics; bx: biopsy; IF: immunofluorescence; CYC: cyclophosphamide; CT: computed tomography; rFVIIa: recombinant activated factor VII; IV: intravenous; UDS: urine drug screen; ACEI: angiotensin-converting enzyme inhibitor: LC-MS: liquid chromatography-mass spectrometry; GN: glomerulonephritis.

**Table 2 pharmaceuticals-16-00846-t002:** Serologic profiles and renal indices at the time of pulmonary-renal syndrome presentation in cases of pulmonary-renal syndrome secondary to levamisole-adulterated cocaine-induced systemic antineutrophil cytoplasmic antibody (ANCA)-associated vasculitis identified through literature review.

Reference	Age (Years)	Sex	Race, Ethnicity	C-ANCA	P-ANCA	PR3 Ab	MPO Ab	APLA	ANA	Other Abs	Cr (mg/dL)	Urine Protein	Urine Blood	Urine Microscopy
Neynaber et al., 2008 [30]	22	M	NR, NR	pos 1:640	neg	pos	neg	NR	neg	NR	1.5	pos (nephrotic range)	pos	NR
Pecci et al., 2013 [29]	37	M	NR, NR	NR	NR	NR	pos	neg (NOS which Abs)	NR	NR	“Normal”	pos	pos	NR
Carlson et al., 2014 [26]	49	F	NR, NR	neg	pos 1:5120	neg	pos	Cardiolipin IgM: pos LAC: pos	neg	NR	7.31	pos (nephrotic range)	pos	Dysmorphic RBCs
Collister et al., 2017 [27]	35	M	White, NR	neg	pos	24 RU/mL *	pos	NR	neg	GBM Ab: neg Cryo: neg dsDNA: 9 IU/mL * RF: 11 IU/mL *	1.7	pos (nephrotic range)	pos	Dysmorphic RBCs
Berlioz et al., 2017 [25]	41	F	NR, Hispanic	neg	pos 1:5120	neg	pos	Cardiolipin IgG: neg Cardiolipin IgM: neg Beta-2-glycoprotein: neg (Ig not specified)	pos 1:160, speckled	dsDNA: pos 1:160 RF: neg Anti-SMA: neg Anti-CCP: neg	6.3	pos	pos	NR
Habibullah, Lou, & Tsegaye, 2019 [28]	53	F	NR, NR	neg	pos 1:80	neg	neg	NR	NR	NR	“Acute renal failure”	NR	NR	NR
Restrepo-Escobar et al., 2020 [31]	34	M	NR, NR	NR	NR	pos	pos	Cardiolipin IgM: pos Cardiolipin IgG: neg LAC: pos Beta-2-glycoprotein-1 IgG: neg Beta-2-glycoprotein-1 IgM: neg	pos 1:160, spotted	GBM Ab: neg Anti-Ro: pos Anti-La: pos Anti-Sm: neg Anti-RNP: neg RF: pos	2.15	pos (non-nephrotic range)	pos	70% crenated RBCs
Voong & Patel, 2022 [32]	58	F	NR, NR	NR	NR	NR	NR	NR	NR	NR	11.07	NR	NR	NR

M: male; F: female; C-ANCA: cytoplasmic antineutrophil cytoplasmic antibody; P-ANCA: perinuclear antineutrophil cytoplasmic antibody; Ab: Antibody; PR3: proteinase-3; MPO:myeloperoxidase: APLA: antiphospholipid antibodies; ANA: Antinuclear antibody; Cr: Creatinine; NR: Not reported; pos: positive; neg: negative; GBM: glomerular basement membrane; NOS: Not otherwise specified; Cryo: Cryoglobulins; LAC: lupus anticoagulant; RBC: red blood cells; dsDNA: double-stranded DNA; RF: rheumatoid factor; Anti-SMA: anti-smooth muscle antibody; anti-CCP: anti-cyclic citrullinated peptide; Anti-Sm: Anti-Smith; Anti-RNP: Anti-ribonucleoprotein; * Normal range for laboratory parameter not reported.

## Data Availability

Data sharing not applicable.

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
