# Peer review of "Pulmonary-Renal Syndrome from Levamisole-Adulterated Cocaine-Induced Antineutrophil Cytoplasmic Antibody (ANCA)-Associated Vasculitis: A Systematic Review"

_pharmaceuticals, 2023, doi:10.3390/ph16060846_

Round 1

Reviewer 1 Report

Interesting and actual review. Lungs and kidneys damage (together or in isolation) are quite common in systemic vasculitis, but practicing rheumatologists rarely associate them with cocaine or levamisole. The reason for this is that in the minds of doctors, cocaine is not usually associate with additional exposure to levamisole. Determination of levamisole in biological substrates for suspected cocaine use is also not a routine practice. This review highlights the importance of a more thorough history taking to identify cocaine use in in patients with systemic vasculitis and pulmonary-renal syndrome.  A therapeutic strategy based on avoiding cocaine use may be an effective treatment in this case.

However, there are a number of inaccuracy in the review that need to be corrected.

First, in the introduction by the author, it is necessary to differentiate in more detail the pathology associated with the use of cocaine itself and cocaine with the addition of levamisole. So, most of the problems of ENT organs (sinusitis, destruction of the nasal septum, epitaxies, etc.) can also occur when using only cocaine. Therefore, it is necessary either not to mention them, or to show that their cause is vasculitis caused precisely by levamisole. In addition, evidence must be provided that cocaine itself does not cause damage to the lungs and kidneys. Also it need to provide evidence that levamisole itself (without cocaine) can cause pulmonary-renal syndrome. The article mentions that it can cause convulsions, agranulocytosis and vasculitis, but does not provide evidence that vasculitis can cause damage to the lungs and kidneys.

In addition, I would like to see a more detailed description of the mechanism of the toxic effect of levamisole on the immune system and the mechanisms of development of vasculitis during its use, even if these studies were performed on animal models. After answers to these remarks the article can be published.

Reviewer 2 Report

this is a summary of all case reports of levamisole-cocaine induced pulmonary renal syndromes.  It is well written, and a nice summary of this very rare issue.  The findings are not novel, but it is a worthwhile summary of the small number of cases of this issue

I have no concerns and needs no modifications

Reviewer 3 Report

This review manuscript covered pulmonary-renal syndrome (PRS) in levamisole-adulterated cocaine (LAC) induced AAVs and their various treatment. It showed the proofs-of-concept through various examples of studies. Thus, it is publishable in this journal with minor revisions.  One concern is the title which seems like a research manuscript. Instead, it will be better like “Literature review of pulmonary-renal ~~~”.

This review manuscript covered pulmonary-renal syndrome (PRS) in levamisole-adulterated cocaine (LAC) induced AAVs and their various treatment. It showed the proofs-of-concept through various examples of studies. Thus, it is publishable in this journal with minor revisions.  One concern is the title which seems like a research manuscript. Instead, it will be better like “Literature review of pulmonary-renal ~~~”.
